# Reciprocal Regulation Between Indoleamine 2,3-Dioxigenase 1 and Notch1 Involved in Radiation Response of Cervical Cancer Stem Cells

**DOI:** 10.3390/cancers12061547

**Published:** 2020-06-12

**Authors:** Hui-Ying Low, Yueh-Chun Lee, Yi-Ju Lee, Hui-Lin Wang, Yu-I Chen, Peng-Ju Chien, Shao-Ti Li, Wen-Wei Chang

**Affiliations:** 1Institute of Medicine, Chung Shan Medical University, Taichung 40201, Taiwan; huiying716@gmail.com (H.-Y.L.); yijulee@csmu.edu.tw (Y.-J.L.); 2Department of Radiation Oncology, Chung Shan Medical University Hospital, Taichung 40201, Taiwan; lee.yuehchun@gmail.com (Y.-C.L.); showtear@gmail.com (S.-T.L.); 3School of Medicine, Chung Shan Medical University, Taichung 40201, Taiwan; 4Department of Biomedical Sciences, Chung Shan Medical University, Taichung 40201, Taiwan; huilin117@gmail.com (H.-L.W.); leatw170@yahoo.com.tw (Y.-I.C.); chienpengju@gmail.com (P.-J.C.); 5Department of Medical Research, Chung Shan Medical University Hospital, Taichung 40201, Taiwan

**Keywords:** IDO1, notch1, radiation, cervical cancer, cancer stem cells

## Abstract

Cervical cancer is the fourth most common cancer in women around the world. Cancer stem cells (CSCs) are responsible for cancer initiation, as well as resistance to radiation therapy, and are considered as the effective target of cancer therapy. Indoleamine 2,3-dioxygenase 1 (IDO1) mediates tryptophan metabolism and T cell suppression, but the immune-independent function of IDO1 in cancer behavior is not fully understood. Using tumorsphere cultivation for enriched CSCs, we firstly found that IDO1 was increased in HeLa and SiHa cervical cancer cells and in these two cell lines after radiation treatment. The radiosensitivity of HeLa and SiHa tumorsphere cells was increased after the inhibition of IDO1 through RNA interference or by the treatment of INCB-024360, an IDO1 inhibitor. With the treatment of kynurenine, the first breakdown product of the IDO1-mediated tryptophan metabolism, the radiosensitivity of HeLa and SiHa cells decreased. The inhibition of Notch1 by shRNA downregulated IDO1 expression in cervical CSCs and the binding of the intracellular domain of Notch (NICD) on the IDO1 promoter was reduced by Ro-4929097, a γ-secretase inhibitor. Moreover, the knockdown of IDO1 also decreased NICD expression in cervical CSCs, which was correlated with the reduced binding of aryl hydrocarbon receptor nuclear translocator to Notch1 promoter. In vivo treatment of INCB-0234360 sensitized SiHa xenograft tumors to radiation treatment in nude mice through increased DNA damage. Furthermore, kynurenine increased the tumorsphere formation capability and the expression of cancer stemness genes including Oct4 and Sox2. Our data provide a reciprocal regulation mechanism between IDO1 and Notch1 expression in cervical cancer cells and suggest that the IDO1 inhibitors may potentially be used as radiosensitizers.

## 1. Introduction

Cervical cancer is the fourth most common cancer in women, accounting for 6.6% of all cancers in women and with an estimated 570,000 new cases and 311,000 deaths in 2018 [1]. Persistent human papillomavirus (HPV)-16 and -18 infection is the most important factor in cervical cancers [2] and the progenitor cells located in the squamocolumnar junction of cervical tissue being infected by oncogenic HPV are more easily to transform into cervical intraepithelial neoplasia [3]—such observations fit the concept of cancer stem cells (CSCs). CSCs are a subpopulation of cancer cells that participate in cancer initiation [4,5], drug resistance [6,7] and metastasis [8,9], and these cells are considered as the most important cancer cells to be elucidated for creating a successful cancer therapy. Current treatments for cervical cancer include surgery, chemotherapy and radiation therapy; however, the existence of cervical CSCs has been considered as one of the reasons for radioresistance [10].

Indoleamine 2,3-dioxygenase 1 (IDO1) is an enzyme responsible for the oxidative metabolism of tryptophan, which converts tryptophan to kynurenine and inhibits the activation and proliferation of T cells [11]. IDO1 was found to be overexpressed in certain types of cancer and causes immune escape and metastasis, resulting in poor prognosis of patients [12]. It has been reported that cervical cancer cells with IDO1 expression were mostly found in cells at the invasive front [13] and the high ratio of kynurenine/tryptophan was positively correlated with lymph node metastasis, tumor size and low disease-free survival of cervical cancer patients [14], suggesting that IDO1 may be involved in the progression of cervical cancers. In addition to the T cell suppression activity, IDO1 has been reported to support cancer growth in a non-immunomodulation way. Maleki Vareki S. et al. found that the inhibition of IDO1 expression in human lung cancer cells sensitized cells to chemotherapeutic agents through the regulation of the cell cycle [15]. In colorectal cancer cells, it was found that tryptophan metabolites, kynurenine or quinolinic acid, could activate β-catenin pathway and promote the cell proliferation [16]. Although the non-immune functions of IDO1 have been reported in some cancer types, the involvement of IDO1 in the characteristics of cervical CSCs remains unclear.

In the present study, we firstly found that the protein expression of IDO1 was increased in tumorspheres derived from HeLa and SiHa cervical cancer cells, and in cells after irradiation. The inhibition of IDO1 by RNA interference or the treatment of a small molecule inhibitor sensitized the cervical tumorsphere cells to radiation treatment. The elevated IDO1 expression in cervical tumorsphere cells was inhibited by the inactivation of Notch1 activity by the knockdown of Notch1 or the treatment of a γ-secretase inhibitor. We further discovered that the reduction of IDO1 expression also inhibited Notch1 activation and the addition of kynurenine, the first breakdown product of IDO1 mediated tryptophan metabolism, increased the self-renewal capability of cervical cancer cells. In addition, the treatment of INCB-024360 before radiation treatment enhanced the efficacy of radiotherapy in vivo through the increased DNA damage. Our data suggest a reciprocal regulation mechanism between IDO1 and Notch1 in cervical CSCs which leads to their radioresistance feature.

## 2. Results

### 2.1. IDO1 is Upregulated in Cervical CSCs and Cerivcal Cancer Cells after Exposure of Radiation

Using the Gene Expression Profiling Interactive Analysis webtool (GEPIA, http://gepia.cancer-pku.cn/index.html) to analyze the data from the Cancer Genome Atlas (TCGA), the expression of *IDO1* mRNA was significantly increased in cervical cancer tissues when compared to normal cervical tissues (Figure 1A). To explore the potential function of IDO1 in CSC behavior, we firstly examined its expression between two cultivation methods of conventional two-dimensional (2D) and tumorsphere, a cell culture based method to enrich CSCs [17,18], and found that IDO1 protein expression was upregulated in cervical tumorspheres from HeLa and SiHa cervical cancer cells (Figure 1B). In addition to protein level, the IDO1 activity, which was determined by the conversion of kynurenine from tryptophan using Ehlrich reagent, was also increased in HeLa and SiHa tumorspheres in comparison to 2D cultured cells (Figure 1C). Radioresistance is one of the features of CSCs, including cervical cancers [10,19]. We examined the expression of IDO1 in HeLa and SiHa cells after 2 Gy radiation treatment and the results showed that the IDO1 protein level increased by radiation stimulation (Figure 1D). The suppressive effect to the proliferation of Jurkat T cells with the conditional media collected from irradiated HeLa and SiHa cells was significantly enhanced in comparison to non-irradiated cells (Figure 1E), supporting the observations of the IDO1 upregulation in irradiated cervical cancer cells. These data clearly demonstrate that IDO1 activity is elevated in cervical CSCs or irradiated cervical cells and also suggests that IDO1 activity may be involved in the radiation response of cervical CSCs.

### 2.2. Inhibition of IDO1 Decreases Radiosensitivity of Cervical CSCs

To explore the involvement of IDO1 in the radiosensitivity of cervical CSCs, we collected tumorspheres from HeLa and SiHa cells and determined their radiosensitivity after the knockdown of IDO1. The colony number of IDO1 knockdown cervical tumorsphere cells from HeLa and SiHa cells was significantly decreased as radiation increased when compared to the control shRNA transduced cells (Figure 2A). The sensitizer enhancement ratios (SERs) for an estimated fractional survival (FS) as 0.5 of IDO1 knockdown cells were 2.41 or 1.43 for two sh-IDO1 transduced HeLa tumorsphere cells and 1.70 or 1.30 for two sh-IDO1 knockdown SiHa tumorsphere cells. These results revealed that the knockdown of IDO1 sensitized the cervical CSCs to radiation treatment. In addition to the RNA interference method, we also used INCB-024360, a small molecule inhibitor of IDO1 with a high specificity and a lead agent for clinical evaluations [20] to inhibit IDO1 activity, followed by the determination of radiosensitivity. Similar to the RNA interference data, the treatment of INCB-024360 in HeLa and SiHa tumorsphere cells increased their radiosensitivity as the colony numbers were significantly decreased in a dose dependent manner (Figure 2B). The SERs of FS at 0.5 were increased in a dose-dependent manner in both HeLa and SiHa tumorsphere cells (Figure 2B). These results indicate that the increased IDO1 activity in cervical CSCs protects cells from radiation treatment.

### 2.3. Notch1 Activation Contributes to the Increased IDO1 Expression in Cervical CSCs

It is known that the activation of Notch1 contributes to the radioresistance in several cancer types [21,22,23,24]. We firstly observed that the expression of the intracellular domain of Notch (NICD), the activation form of Notch1, was increased in tumorspheres derived from HeLa and SiHa cells (Figure 3A). We next examine the role of Notch1 in IDO1 expression of cervical CSCs. With lentiviral delivery of Notch1 specific shRNAs to inhibit NICD expression, the IDO1 protein level was downregulated (Figure 3B). We also found that the kynurenine concentration in culture supernatant of HeLa and SiHa tumorspheres was significantly reduced after knockdown of Notch1 (Figure 3C). In addition to RNA interference, we also used Ro-4929097, a γ-secretase inhibitor, to inhibit Notch1 activation in HeLa and SiHa tumorspheres and found that the IDO1 expression was also downregulated (Figure 3D). The activated NICD is known to translocate into the nucleus to turn on the downstream target genes through the association with RBPJ (the recombination signal binding protein for immunoglobulin kappa J region)/CSL (CBF1/Suppressor of Hairless/LAG-1) transcription factor complex [25]. From the analysis of the IDO1 promoter on the Eukaryotic Promoter Database (EPD, https://epd.epfl.ch//index.php), there was a putative RBPJ/CSL binding site of (-TGTGGGAA-) at –790 upstream or +140 downstream from the transcription start site (TSS) (Figure 3E). With the chromatin immunoprecipitation (ChIP) method using an anti-Notch1 antibody to pull-down chromatins, followed by the detection of the IDO1 promoter with quantitative polymerase chain reaction (qPCR) analysis, we further found that the treatment of Ro-4929097 strongly suppressed the binding of NICD to the IDO1 promoter at the +140 site, one of the putative RBPJ/CSL binding sites, in both HeLa and SiHa tumorspheres (Figure 3E). We also confirmed that the knockdown of Notch1 in HeLa and SiHa tumorspheres increased their sensitivity to radiation treatment (Figure 4). The SERs for a FS of 0.5 in Notch1 knockdown cells were 2.19 and 1.48 for HeLa tumorsphere cells and 1.53 and 1.83 for SiHa tumorsphere cells (Figure 4). These results suggest that Notch1 activation positively regulates the upregulation of IDO1 in cervical CSCs.

### 2.4. IDO1 also Regulates Notch1 Expression in Cervical CSCs

Liu et al. recently reported that IDO1 mediated the maintenance of pluripotency in human embryonic stem cells [26] and led us to investigate the potential regulatory activity of IDO1 in Notch1 expression. With the lentiviral delivery of IDO1 specific shRNAs (Figure 5A) or the treatment of INCB-024360 (Figure 5B), the inhibition of IDO1 in HeLa and SiHa tumorspheres caused the downregulation of NICD. It is known that kynurenine is an endogenous ligand for the aryl hydrocarbon receptor (AhR) [27]. The binding of kynurenine and AhR further interacts with the aryl hydrocarbon receptor nuclear translocator (ARNT) to form a complex to the aryl hydrocarbon response element and initiate the transcription of target genes [28]. After the analysis of Notch1 promoter on EPD website, there were four putative AhR/ARNT binding sites of (-GCGTG-) at −17, −309, −395, and −924 upstream from TSS (Figure 5C). Using an anti-ARNT antibody to perform ChIP analysis, we further demonstrated that the knockdown of IDO1 reduced the binding of ARNT to Notch1 promoter at the −309 site in the HeLa and SiHa tumorsphere cells (Figure 5C). These data indicate that there is a reciprocal regulation mechanism between IDO1 and Notch1 in cervical CSCs.

### 2.5. INCB-024360 Could Serve as a Radiosensitizer In Vivo

Due to the observations that the inhibition of IDO1 activity enhanced the radiosensitivity of cervical CSCs (Figure 2), we hypothesize that the IDO1 inhibitors could function as radiosensitizers for helping the radiation therapy in cervical cancer. After subcutaneous injection of SiHa tumorsphere cells into the back skin of nude mice to form xenograft tumors, we examined the radiosensitizer potential of INCB-024360 with an injection dose of 50 mg/kg at the day before radiation treatment. The final tumor sizes and weights in the group of INCB-024360 pre-injection plus radiation were smaller than the radiation alone group (Figure 6A,B). The expression of Ki-67, the proliferation marker, was significantly decreased, whereas the expression of phosphor-γH2Ax^ser139^, the marker for DNA damage, was significantly increased in the group of INCB-024360 pre-injection plus radiation when compared to the radiation alone group (Figure 6C,D). These results suggest that INCB-024360 is a potential radiosensitizer to cervical cancer.

### 2.6. IDO1 and Kynurenine Supports Self-renewal of Cervical CSCs

Due to the observation of the trend of slowed tumor growth (Figure 6A) and the decreased Ki-67 expression in INCB-024350 treated SiHa xenograft tumors (Figure 6D), we hypothesize that IDO1 expression contributes to the maintenance of cervical CSCs. With tumorsphere assay, the self-renewal capability of SiHa cells, which was determined by the formation of secondary tumorspheres, was found to be decreased after the knockdown of IDO1 (Figure 7A). Moreover, the addition of kynurenine to tumorsphere cultivation significantly increased the tumorsphere number of HeLa and SiHa cells (Figure 7B), as well as the increased expression of Oct4 or Sox2 but not BMI1 (Figure 7C), which all belong to the well-known cancer stemness genes [29]. These results indicate that IDO1 and kynurenine, the main tryptophan metabolite, support the self-renewal of cervical CSCs.

## 3. Discussion

In the present study, we discovered that IDO1 expression was upregulated in cervical CSCs to decrease their radiosensitivity (Figure 1A and Figure 2). These findings were consistent with a recent report by Chen et al. that the inhibition of IDO1 sensitized colorectal cancer cells to radiation-induced cell death [30]. Li et al. found that IDO1+ myeloid suppressor cells were elevated in Lewis lung tumors in C57BL/6 mice after three applications of 12 Gy radiation, which was called radiation-induced rebound immune suppression [31]. The treatment of INCB-023843, another small molecule inhibitor of IDO1, could overcome radiation-induced immune suppression in the mouse Lewis lung cancer model [31]. Similar results were also observed in glioblastoma. Kesarwani et al. found that kynurenine was accumulated in human glioblastoma tumors and the radiation-induced regulatory T cells in a mouse model of glioblastoma could be mitigated by the treatment of an IDO1 inhibitor, GDC-0919 [32]. The results from a phase I clinical trial of INCB-024360 advanced solid tumors indicated the monotherapy of INCB-024360 did not show the significant anti-tumor activity [33]. The growth inhibition of SiHa xenograft tumors in our present study also did not reach to a significant reduction (Figure 6A,B). It suggests that the combination of INCB-024360 with other treatment, such as radiation or immune checkpoint inhibitors, may be required to obtain the tumor regressions, which could be observed in our data of SiHa xenografts after the combination of INCB-024360 and radiation (Figure 6). 

We further identified that the increased IDO1 expression in cervical CSCs from HeLa and SiHa cells was mediated by the activation of Notch1 (Figure 3). Using the GEPIA webtool to analyze the RNA sequencing expression data from the TCGA database, there was a significantly positive correlation between IDO1 and RBPJ (R = 0.25, *p* = 1.3 × 10^−5^, Appendix A), which further supported our results from the cell experiments. The positive correlation between IDO1 and Notch1 from GEPIA analysis was also observed (R = 0.17, *p* = 0.0035, Appendix A), which also supported the presence of reciprocal regulation between IDO1 and Notch1 in the clinical specimens from the TCGA database. Our data revealed that the binding of ARNT to Notch1 promoter was reduced in HeLa and SiHa tumorsphere cells after IDO1 knockdown (Figure 5C). The exogenous kynurenine supplement also displayed an enhancement in the tumorsphere formation of HeLa and SiHa cells (Figure 7B). The analysis of the TCGA data by the GEPIA webtool revealed the strong positive correlation between AhR and Notch1 (R = 0.39, *p* = 2.5 × 10^−12^, Appendix A) and between ARNT and Notch1 (R = 0.54, *p* = 0, Appendix A) in cervical cancer patients. Thaker et al. demonstrated that the exogenous administration of kynurenine or quinolinic acid caused the activation of β-catenin and induced proliferation and tumor growth of human colon cancer cells [16]. Using the SurvExpress webtool (http://bioinformatica.mty.itesm.mx:8080/Biomatec/SurvivaX.jsp) to analyze the TCGA data, both the AHR and CTNNB1 expression level was significantly higher in the high risk group of cervical cancer patients, and the combination of AHR/CTNNB1 could serve as a poor prognostic factor in cervical cancer (Appendix A). These data suggest that the kynurenine/AhR/ARNT pathway may contribute to the progression of cervical cancer. 

We demonstrated that kynurenine could enhance self-renewal capability of HeLa and SiHa cells (Figure 7B) with the upregulation of Oct4 or Sox2 expression (Figure 7C). Venkateswaran et al. recently reported that only kynurenine, not other tryptophan metabolites, promoted the nuclear translocation of AhR in colon cancer cells [34]. Blocking the kynurenine/AhR interaction by CH-223191, an AhR antagonist, inhibited the proliferation of colon cancer cells [34]. It is worth investigating the CSC inhibition potential of AhR antagonists in cervical cancer. Wang et al. reported that the overexpression of a truncated form of AhR in HeLa cells greatly reduced their tumorigenicity in vivo and the degradation of ARNT was observed [35]. Using the GEPIA to analyze the TCGA data, the higher expression level of AhR was positively correlated with shorter disease-free survival of cervical cancer patients (Appendix A). This suggests that the inhibition of AhR/ARNT activity is potentially developed as a therapeutic strategy for cervical cancer.

## 4. Materials and Methods 

### 4.1. Chemicals

The IDO1 inhibitor, INCB-024360, and the γ-secretase inhibitor, Ro-4929097, were purchased from Cayman Chemical Company (Ann Arbor, MI, USA) and dissolved in dimethyl sulfoxide (Sigma-Aldrich, St. Louis, MO, USA) as a stock of 50 mM and store at −20 °C. L-kynurenine was purchased from Sigma-Aldrich and was dissolved in 0.5M HCl as a stock of 50 mM and store at −20 °C.

### 4.2. Cell Culture and Tumorsphere Cultivation

HeLa and SiHa cells were provided from Professor Jiunn-Liang Ko (Institute of Medicine, Chung Shan Medical University, Taichung, Taiwan) and the cell identity was confirmed using short tandem repeat analysis, which was carried out by the Center of Genomic Medicine at National Cheng Kung University (Tainan, Taiwan). Cells were maintained by Dulbecco’s Modified Eagle Medium (DMEM) (Gibco, Thermo Fisher Scientific, Waltham, MA, USA) with 10% fetal bovine serum (Biological Industries, Beit-Haemek, Israel.), 1 mM sodium pyruvate (Biological Industries), and 1X penicillin-streptomycin (Biological Industries) in a 5% CO_2_ air humidified atmosphere at 37 °C. For primary tumorsphere cultivation, cells were harvested and suspended at 2 × 10^3^ cells/well in DMEM/F12 medium containing 0.4% bovine serum albumin (Gibco), 20ng/mL epidermal growth factor (PeproTech Asia, Rehovot, Israel), 20ng/mL basic fibroblast growth factor (PeproTech), 5 μg/mL insulin, 0.5X B27 supplement (Invitrogen, Thermo Fisher Scientific, Waltham, MA, USA), and 4μg/mL heparin (Sigma-Aldrich) in a suspension 6-well-plate (Greiner Bio-One GmbH, Kremsmünster, Austria) at 37 °C incubator for 7 days. For secondary tumorsphere cultivation, primary tumorspheres were harvested by a 100 μm cell strainer (BD Biosciences, San Jose, CA, USA) and dissociated into single cells by HyQTase (Hyclone Laboratories, Inc., Logan, UT, USA) treatment at 37 °C for 3–5 min followed by being suspended in tumorsphere media at 1 × 103 cells/well and cultured as primary tumorsphere cultivation as described above.

### 4.3. Radiation Treatment and Clonogenic Assay of Cervical Cancer Cells 

Cells were seeded onto 6 cm cell culture dishes with 4 mL culture media and performed radiation as the indicated dose with Elekta Axesse^TM^ linear accelerator (Elekta AB, Stockholm, Sweden) at a dose rate as 6 Gy min^−1^. After irradiation, the dishes were then changed with fresh media and cultured at 37 °C for further experiments. For the clonogenic assays, the cells were seeded in a 6-well-plate at a density of 500 cells/well in culture media and performed irradiation as described above and cultured for colony formation for 14 days. The colonies formed were stained by 0.01% (w/v) crystal violet in 3.7% formaldehyde/phosphate buffered saline (PBS) solution at room temperature for 30 min followed by ddH2O wash. The colony number was counted by ImageJ software (version 1.8.0, NIH, Bethesda, MD, USA). The curves of fractional survival were calculated and drawn by GraphPad Prism software (version 5.0, GraphPad Software, San Diego, CA, USA) using the linear-quadratic model as the suggestion from the report from Brenner, D.J. [36] The sensitization effects of gene silencing of IDO1 or Notch1 or the treatment of INCB-024360 to cells were evaluated by the calculation of sensitizer enhancement ratios as the report from Naumann et al [37].

### 4.4. Western Blotting

Total cellular proteins were harvested after lysis with a radioimmunoprecipitation assay (RIPA) buffer (GeneTex Inc., Hsinchu City, Taiwan) containing 1X protease/phosphatase inhibitor cocktail (Merck Millipore, Danvers, MA, USA) followed by quantitating protein concentration by BCA reagent (Thermo Fisher Scientific, Waltham, MA, USA). An amount of 25 μg of the total cellular proteins was separated by sodium dodecyl sulfate polyacrylamide gel electrophoresis (SDS-PAGE) and transferred onto polyvinylidene difluoride (PVDF) membrane (Immobilon-P, Merck Millipore, Danvers, MA, USA) followed by blocking with 1% skimmed milk in TBS-T buffer. The membranes were then incubated with primary antibody at 4 °C overnight and peroxidase-conjugated secondary antibody at room temperature for 1 h. The signals were developed by chemiluminescence substrate (PerkinElmer Inc., Waltham, MA, USA) and captured with the Luminescence-Image Analyzer (FUSION SOLO, Vilber Lourmat Deutschland GmbH, Germany). The primary antibodies used in this study were listed: rat monoclonal anti-IDO1 (sc-53978, Santa Cruz Biotechnology, Inc., Dallas, TX, USA), rabbit polyclonal anti-Notch1 (ab27526, Abcam, Cambridge, UK), rabbit polyclonal anti-GAPDH (GTX100118, GeneTex Inc., Hsinchu, Taiwan), rabbit polyclonal anti-α-tubulin (GTX112141, GeneTex Inc., Hsinchu, Taiwan). The quantification of band intensity was performed by ImageJ software.

### 4.5. IDO Activity Assay and Determination of Kynurenine Concentration in Cell Culture Supernatant

For the determination of cellular IDO1 activity, 50 μg of total cellular proteins were added into IDO assay buffer (PBS containing 20 mM sodium ascorbate, 10 μM methylene blue, 100 μg/mL catalase, and 400 μM L-Tryptophan (all purchased from Sigma-Aldrich) and incubated at 37 °C for 30 min, followed by adding 100 μL 30% trichloroacetic acid (Sigma-Aldrich) and incubation at 52 °C for 30 min. After adding 200 μL Ehrlich’s reagent (Sigma-Aldrich) and incubating at room temperature at dark for 30 min, 100 μL of the mixture was then transferred into wells of 96-well-plate and measuring the absorbance at 492 nm. For the determination of kynurenine in culture supernatant, 30 μL of 30% trichloroacetic acid was added into 60 μL of collected supernatant or the kynurenine standards and incubated at 52 °C for 30 min. The mixtures were then added with 90 μL of Ehrlich’s reagent and incubated at room temperature in the dark for 30 min, followed by measuring the absorbance at 492 nm.

### 4.6. T Cell Inhibition Assay

Jurkat T cells were obtained from Dr. Chia-Ling Chen (Department of Respiratory Therapy, School of Medicine, College of Medicine, Taipei Medical University, Taipei, Taiwan) and used as a model for the analysis of the anti-T cell proliferation effect of the cancer cell culture supernatant. Briefly, culture supernatant from cancer cells or tumorspheres was collected and mixed with RPMI-1640/10 % FBS as a ratio of 1:1 and used for Jurkat T cell cultivation. The mixture of fresh media for cancer cell or tumorsphere cultivation with RPMI-1640/10% FBS were used as a control. The cell proliferation was then determined at day 3 by adding Cell Counting Kit-8 reagent (Sigma-Aldrich) and measuring the absorbance at 450 nm.

### 4.7. Lentiviral Delivery of shRNAs

IDO1 (TRCN0000056744 or TRCN0000056747), Notch1 (TRCN0000350253 or TRCN0000350330), or LacZ (TRCN0000231722) specific shRNAs were obtained from the National RNAi Core Facility (Academia Sinica, Taipei, Taiwan). Lentivirus production and transduction into cervical cancer cells were performed as our previous report [38]. The successful transduced cells were selected by 2 μg/mL puromycin (TOKU-E, Bellingham, WA, USA) for 48 h.

### 4.8. Chromatin Immunoprecipitation (ChIP)

The cellular chromatin was cross-linked in cells by adding formaldehyde as a final concentration of 1% at room temperature for 10 min followed glycine quenching. Cells were then suspended in cold PBS containing protease inhibitor cocktail and performed chromatin shearing by sonication. An amount of 25 μg of sheared chromatin DNA was used for ChIP as the protocol described in our previous report [38]. The antibodies used for ChIP in this study were listed: rabbit anti-Notch1 (ab27526, Abcam) and rabbit monoclonal anti-ARNT (#3414, Cell Signaling Technology, Danvers, MA, USA). The pull-down chromatin DNA was extracted using a QIAquick PCR Purification Kit (Qiagen, Venlo, the Netherlands) and quantitated by quantitative PCR with the primer sets listed in Appendix A.

### 4.9. Quantitative RT-PCR (qRT-PCR)

The total RNA was extracted by a Quick-RNA Miniprep Kit (Zymo Research, Irvine, CA, USA) and 1 μg of extracted total RNA was used for the synthesis of complementary DNA (cDNA) by SuperScript™ III First-Strand Synthesis System (Invitrogen). The synthesized cDNA were used as templates for the quantitation of target gene expression by SYBR^TM^ Green Master Mix (Bio-Rad Laboratories, Inc., Hercules, CA, USA) and a PCRmax Eco 48 real-time PCR system (PCRmax, Staffordshire, UK). The sequences of primer sets were listed as Appendix A.

### 4.10. In Vivo Radiation Treatment of SiHa Xenograft Tumors

The animal experiment protocol was reviewed and approved by the Institutional Animal Care and Use Committee (IACUC) in Chung Shan Medical University (IACUC Approval No. 1851). The 1 × 10^5^ SiHa tumorsphere cells were suspended in 0.5 mg/mL matrigel (BD Biosciences) and subcutaneously injected at the back of athymic female nude mice (purchased from the National Laboratory Animal Center, Taipei, Taiwan) for tumor growth. As the tumor volume reached to 50 mm^3^, the mice were divided into four groups: (1) no treatment, (2) INCB-024360 treatment, (3) radiotherapy, (4) INCB-024360 plus radiotherapy. INCB-024360 treatment was performed by intraperitoneal injection of a single dose of 50 mg/kg, the dosage was as Koblish et al. have reported [39]. Radiotherapy was performed by 2 Gy per day for a total dose of 10 Gy. The mice with radiotherapy were orally fed with glutamine solution, beginning from the first day of radiation treatment and lasting for one month. All the mice were then sacrificed and the xenografted tumors were taken out for weighting and embedded into paraffin for immunohistochemistrical analysis.

### 4.11. Immunohistochemistrical Analysis

The paraffin-embedded SiHa xenografted tumors were sliced into 4 μm and deparaffinization performed according to our previous reports [38,40]. The cell proliferation or DNA damage within tumor tissues was determined by the expression of Ki-67 or phosphor-γH2AX^ser139^ using the corresponding antibody as rabbit monoclonal anti-Ki-67 (GTX16667, GeneTex Inc., Hsinchu, Taiwan) or rabbit polyclonal anti-p-γH2AX^ser139^ (NB100-2280, Novus Biologicals, LLC., Centennial, CO, USA). The quantitative results of Ki-67 or p-γH2AX^ser139^ expression were performed by TissueFAX software (TissueGnostics, Vienna, Austria).

### 4.12. Statistical Analysis

Quantitative data were presented as the mean ± SD. The comparisons between two groups were analyzed with Student’s *t*-test. The comparisons among multiple groups (more than two) were analyzed with repeated measure ANOVA followed by Tukey–Kramer’s post hoc test to identify differences among specific groups using GraphPad Prism software.

## 5. Conclusions

In the present study, we have demonstrated that IDO1 expression was upregulated in cervical CSCs or cervical cancer cells after irradiation. The inhibition of IDO1 by RNA interference or INCB-024360 treatment increased the radiosensitivity of cervical CSCs, indicating that IDO1 expression protects cervical CSCs from radiation treatment. We also found that the increased IDO1 in cervical CSCs was mediated by Notch1 activation through the direct binding of NICD to IDO1 promoter. In addition, IDO1 also regulated Notch1 expression through the binding of AhR/ARNT to Notch1 promoter. The treatment of INCB-024360 in SiHa xenograft model significantly enhanced the efficacy of radiotherapy. Furthermore, kynurenine enhanced the self-renewal capability of cervical cancer cells. These results suggest that IDO1 inhibitors can be potentially developed as radiosensitizers for future cervical cancer therapy.

## Figures and Tables

**Figure 1 cancers-12-01547-f001:**
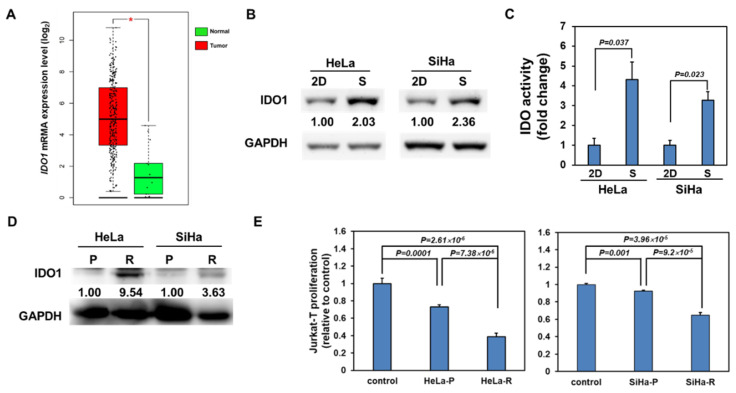
Indoleamine 2,3-dioxygenase 1 (IDO1) is upregulated in cervical cancer stem cells (CSCs) and irradiated cervical cancer cells. (**A**) The expression levels of *IDO1* mRNA among normal cervical or cervical cancer tissues were analyzed by the Gene Expression Profiling Interactive Analysis (GEPIA) website using the data of the Cancer Genome Atlas (TCGA). * *p* < 0.01. (B, C) The total cell proteins were harvested from HeLa and SiHa cells under 2-dimensional culture (2D) or tumorsphere culture (S). The IDO1 protein expression was determined by Western blotting (**B**). The inserted numbers indicated the relative expression level of S in comparison to 2D. The IDO1 activity was determined by the conversion of kynurenine from tryptophan using Ehlrich reagent and measured the absorbance at 492 nm (**C**). The data were presented as fold change to 2D group. (**D**) The total cell proteins were harvested from parental (P) or irradiated (R) HeLa and SiHa cells and the IDO1 protein expression was determined by Western blotting. The inserted numbers indicate the relative expression level of R in comparison to P. (**E**) The culture supernatant of parental (P) or irradiated (R) HeLa and SiHa cells was mixed with RPMI-1640 medium at a ratio of 1:1 and used for examining the T cell suppression effect by measuring the proliferation of Jurkat T cells. The control indicated the mixture of fresh DMEM medium and RPMI-1640 medium.

**Figure 2 cancers-12-01547-f002:**
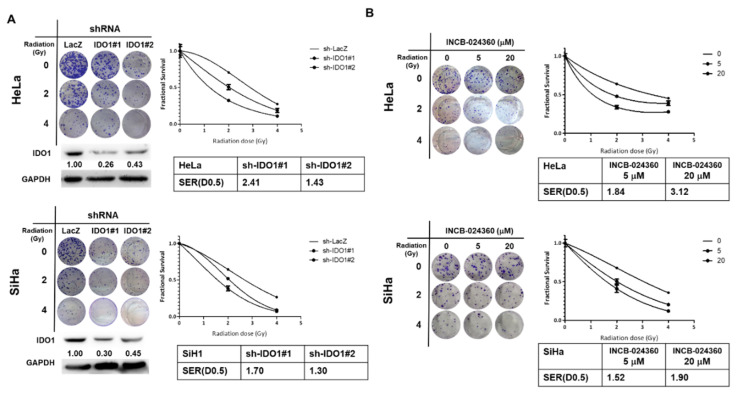
The inhibition of IDO1 sensitizes cervical CSCs to irradiation. Tumorspheres from HeLa and SiHa cells were dissociated into single cell suspension. (**A**) The tumorsphere cells were transduced with lentiviruses carrying IDO1 specific shRNAs (IDO1#1 or IDO1#2) or control LacZ specific shRNA (LacZ) and selected with 2 μg/mL of puromycin. The surviving cells were than used for the examination of radiosensitivity by gamma-irradiation as an indicated dosage followed by clonogenic assay. The sensitizer enhancement ratio (SER) for an estimated fractional survival of 0.5 (D0.5) was calculated as D0.5(sh-LacZ)/D0.5(sh-IDO1). (**B**) The tumorsphere cells were treated with INCB-024360 as the indicated concentration and performed radiation treatment followed by clonogenic assay. 0.1% dimethyl sulfoxide (DMSO) was used as vehicle control. The data were presented as a fractional survival and SER(D0.5) was calculated as D0.5(DMSO)/D0.5(INCB). All the experiments were repeated three times and data from one experiment were presented.

**Figure 3 cancers-12-01547-f003:**
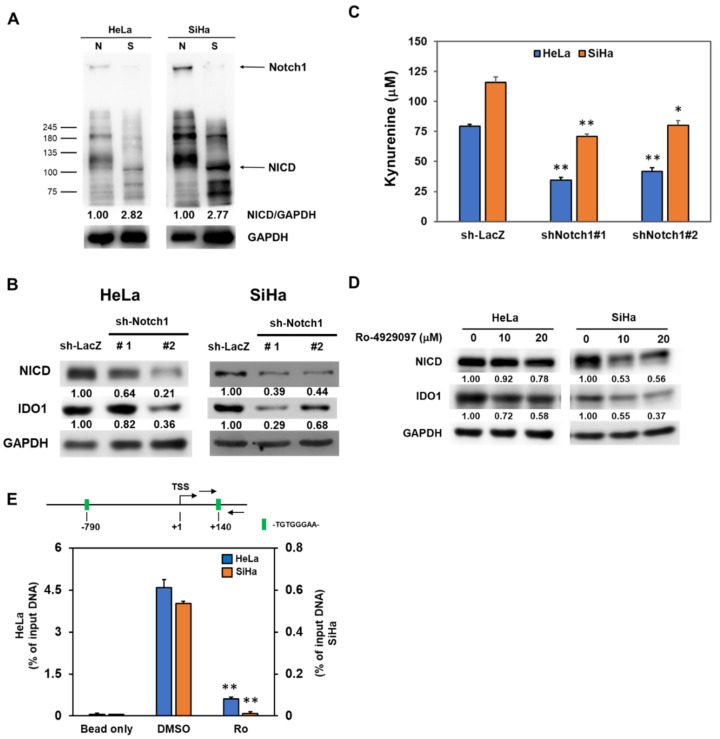
Notch1 activation in cervical CSCs participates to the IDO1 induction. (**A**) Total cell lysates were collected from conventional adherent culture (N) or tumorspheres (S) from HeLa and SiHa cells and the expression of Notch1 and NICD was determined by Western blotting. (**B**–**D**) Tumorspheres from HeLa and SiHa cells were dissociated into single cell suspension and were transduced with lentiviruses carrying Notch1 specific shRNAs (sh-Notch1#1 or sh-Notch1#2) followed by puromycin selection. The surviving cells were collected and detected the expression of NICD or IDO1 by Western blotting (**B**). The insert numbers indicate the relative expression level in comparison to sh-LacZ transduced cells. The kynurenine concentration in the culture supernatant was determined by Ehrlich’s reagent (**C**). * *p* < 0.05; ** *p* < 0.01. Tumorsphere cells were treated with Ro-4929097 as the indicated concentration and detected the expression of NICD or IDO1 by western blot (**D**). The insert numbers indicate the relative expression level in comparison to 0.1% DMSO control (labeled as 0 μM). (**E**) The putative binding site of RBPJ/CSL (-TGTGGGAA-) in IDO1 promoter was analyzed by the Eukaryotic Promoter Database (EPD) website and the binding of NICD to the RBPJ/CSL binding site of IDO1 promoter at +140 site in HeLa and SiHa tumorsphere cells was detected by the chromatin immunoprecipitation (ChIP) method with an anti-Notch1 antibody and quantitated by the qPCR method. Data were presented as percentage of input DNA. ** *p* < 0.01. All the experiments were repeated three times and the data from one experiment were presented.

**Figure 4 cancers-12-01547-f004:**
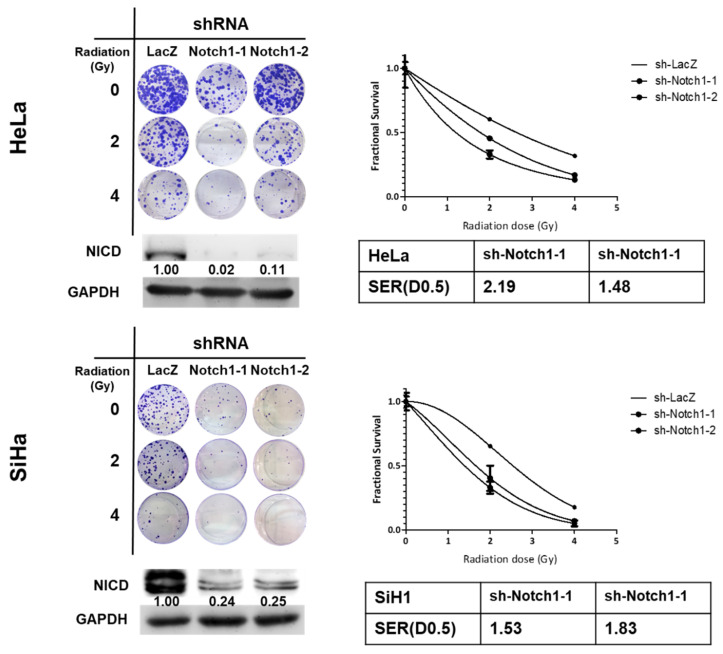
Knockdown of Notch1 in cervical CSCs enhances the efficacy of radiation treatment. HeLa and SiHa tumorsphere cells were transduced with lentiviruses carrying Notch1 specific shRNAs (sh-Notch1-1 or sh-Notch1-2) followed by puromycin selection. The surviving cells were collected and radiation treatment performed, followed by clonogenic assay. The data were presented as fractional survival and the SER for an estimated fractional survival as 0.5 (D0.5) (SER(D0.5)) was calculated by D0.5(sh-LacZ)/D0.5(sh-Notch1). All the experiments were repeated three times and data from one experiment were presented.

**Figure 5 cancers-12-01547-f005:**
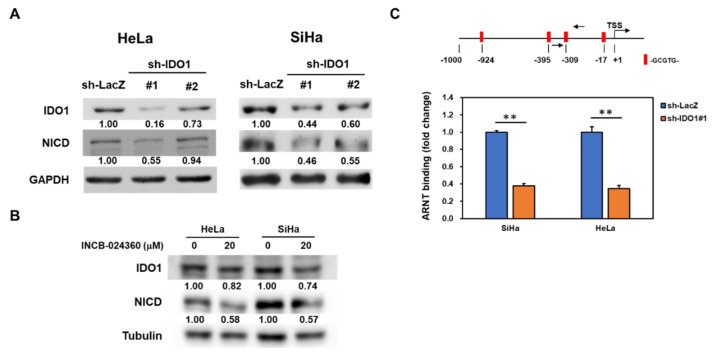
Inhibition of IDO1 in cervical CSCs reduces Notch1 activation. Tumorspheres from HeLa and SiHa cells were dissociated into single cell suspension. (**A**,**B**) The tumorsphere cells were transduced with lentiviruses carrying IDO1 specific shRNAs (IDO1#1 or IDO1#2) or control LacZ specific shRNA (LacZ) followed by selection with puromycin (**A**) or treatment with 20 μM of INCB-024360 for 48 h. The expression of NICD or IDO1 was determined by Western blotting. The insert numbers indicated the relative expression level in comparison to sh-LacZ transduced cells. (**C**) The putative binding sites of aryl hydrocarbon receptor/aryl hydrocarbon receptor nuclear translocator (AhR/ARNT) (-GCGTG-) in the Notch1 promoter were analyzed by the EPD website and the binding of ARNT to −309 site of Notch1 promoter in HeLa and SiHa tumorsphere cells after the knockdown of IDO1 was analyzed by the ChIP method with an anti-ARNT antibody and quantitated by the qPCR method. ** *p* < 0.01. All the experiments were repeated three times and data from one experiment were presented.

**Figure 6 cancers-12-01547-f006:**
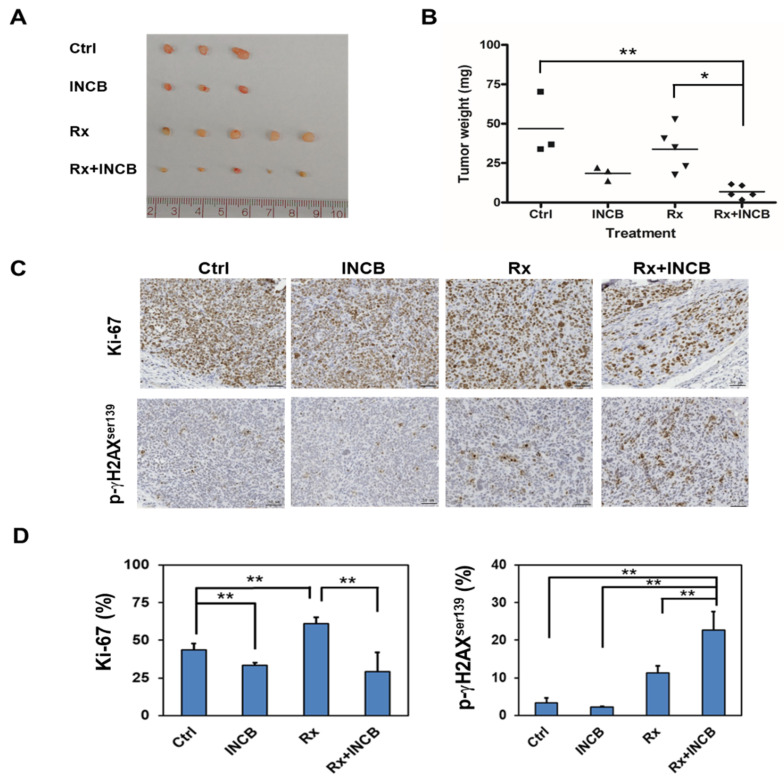
IDO1 inhibitor enhances the efficacy of radiotherapy in vivo. 1 × 10^5^ of SiHa tumorsphere cells were subcutaneously injected to nude mice for tumor growth. After the tumor volume reached 50 mm^3^, the mice were divided into four groups of non-treated (Ctrl), INCB-024360 treated (INCB), radiotherapy (Rx), or INCB-024360 plus radiotherapy (INCB + Rx). For the INCB or INCB + Rx group, mice were injected once with 50 mg/kg INCB-024360 intraperitoneally before radiotherapy. For the Rx or INCB + Rx group, mice received 2 Gy radiation per day for total 10 Gy. Mice were sacrificed at day 30 after the last radiation treatment and the xenografted tumors were taken out for picturing (**A**) and weighting (**B**). The expression of Ki-67 or p-γH2AX^ser139^ was determined by paraffin section followed by immunohistochemical staining (**C**). The inserted bars indicated 50 μM. The quantification results were performed by TissueFAX software (**D**). * *p* < 0.05; ** *p* < 0.01. The experiments were repeated two times and data from one experiment were presented.

**Figure 7 cancers-12-01547-f007:**
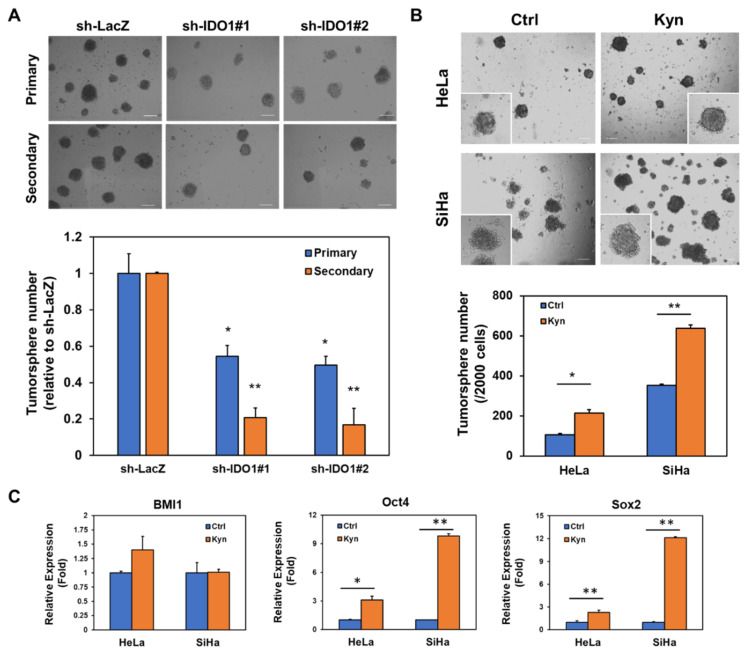
Tryptophan metabolism by IDO1 positively regulates self-renewal of cervical CSCs. (**A**) SiHa cells were transduced with lentiviruses carrying IDO1 specific shRNAs (IDO1#1 or IDO1#2) or control LacZ specific shRNA (LacZ) followed by selection with puromycin. The surviving cells were collected at 48 h and 2 × 10^3^ cells were used to perform primary tumorsphere cultivation and the tumorsphere numbers were counted at day 7, followed by collecting formed primary tumorspheres with 100 μm cell strainer. The collected primary tumorspheres were dissociated into single cells and 1 × 10^3^ cells were used for secondary tumorsphere formation followed by counting the tumorsphere number at day 7. * *p* < 0.05; ** *p* < 0.01 when compared to sh-LacZ group. (**B**,**C**) HeLa and SiHa cells were performed tumorsphere cultivation in the supplement of 100 μM kynurenine. The tumorsphere numbers were counted at day 7. The total RNA was collected at 48 h after kynurenine treatment and the expression of BMI1, Oct4 or Sox2 were determined by qRT-PCR method. * *p* < 0.05; ** *p* < 0.01. All the experiments were repeated three times and data from one experiment were presented.

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
