# Peer review of "Reciprocal Regulation Between Indoleamine 2,3-Dioxigenase 1 and Notch1 Involved in Radiation Response of Cervical Cancer Stem Cells"

_cancers, 2020, doi:10.3390/cancers12061547_

Round 1
Reviewer 1 Report
This article describes a reciprocal regulation mechanism between IDO1 and Notch1 in cervical cancer cells in both adherent state and as tumorspheres. The authors present data indicating that targeting IDO1 by either shRNA or enzymatic inhibition increases tumor cell susceptibility to radiation. These observations were made in tumorsphere culture and recapitulated in vivo. The authors also present data demonstrating a mechanism of this reciprocal regulation via direct interaction with gene promoters. Overall, this research is novel in the context of cervical cancer, and data suggests this is a clinically relevant mechanism. However, some of issues were identified in the data and interpretation.
1. Many of the references in the introduction are to review articles. It is important to reference the primary literature.
2. Interpretation of data from Fig. 1 (Line 96) does not match the data. The only activity demonstrated in this figure is the inhibition of Jurkat cell proliferation, and this cannot be described as immune-independent. Also, there is no data related to radiation at this point in the manuscript.
3. There are potentially significant challenges in colony formation assays as many appear to not start with similar cell numbers, particularly in transduced and selected cells. A difference in starting cell number in untreated groups will impact the data in treated conditions. It is possible that targeting IDO1 expression reduces the plating efficiency; if so, this must be adjusted for to achieve valid data in this assay.
4. In Fig. 3E, it is not surprising that inhibition of Notch1 activation leads to a reduction of NICD binding of any of its promoter binding sites. Perhaps the emphasis should be on the detection of NICD bound to the IDO1 promoter.
5. Also in Fig. 3E and Fig. 5C, there is no rationale for why specific putative binding sites were selected. Data for other sites should be included.
6. The methods do not provide sufficient information to review the number of biological/technical replicates and statistical methods. Methods section should include statement of scientific rigor and statistical methods. Comparison of individual data points in colony formation assays by t test is insufficient. See Franken, N., Rodermond, H., Stap, J. et al. Clonogenic assay of cells in vitro. Nat Protoc 1, 2315–2319 (2006). https://doi.org/10.1038/nprot.2006.339
Author Response
This article describes a reciprocal regulation mechanism between IDO1 and Notch1 in cervical cancer cells in both adherent state and as tumorspheres. The authors present data indicating that targeting IDO1 by either shRNA or enzymatic inhibition increases tumor cell susceptibility to radiation. These observations were made in tumorsphere culture and recapitulated in vivo. The authors also present data demonstrating a mechanism of this reciprocal regulation via direct interaction with gene promoters. Overall, this research is novel in the context of cervical cancer, and data suggests this is a clinically relevant mechanism. However, some of issues were identified in the data and interpretation.
Responses:
We thank the positive feedbacks from the reviewer.
- Many of the references in the introduction are to review articles. It is important to reference the primary literature.
Responses:
We have revised the cited references in the revised manuscript for ref 3-9 and ref 11 by citing the primary literature as suggestion from the reviewer.
- Interpretation of data from Fig. 1 (Line 96) does not match the data. The only activity demonstrated in this figure is the inhibition of Jurkat cell proliferation, and this cannot be described as immune-independent. Also, there is no data related to radiation at this point in the manuscript.
Responses:
We agree with the comments from the reviewer and the descriptions have been revised as “These data clearly demonstrate that IDO1 activity is elevated in cervical CSCs or irradiated cervical cells and it also suggests that IDO1 activity may involve in the radiation response of cervical CSCs.” In the revised manuscript (page 3, line 96-97).
- There are potentially significant challenges in colony formation assays as many appear to not start with similar cell numbers, particularly in transduced and selected cells. A difference in starting cell number in untreated groups will impact the data in treated conditions. It is possible that targeting IDO1 expression reduces the plating efficiency; if so, this must be adjusted for to achieve valid data in this assay.
Responses:
The seeding cell number of the clonogenic assay used in this study was 500 cells. For avoid the possible growth inhibition effect of IDO1 knockdown, the colony number of cells of each group (sh-LacZ, sh-IDO1#1, or sh-IDO1#2) without radiation treatment was used for calculation the survival rate of shRNA transduced cells. We also re-calculated the fractional survival with the linear-quadratic model using GraphPad Prism software (version 5.0, GraphPad Software, San Diego, CA, USA) and calculated the sensitizer enhancement ratios as the report from Naumann et al (PloS one 2017, 12, e0180940). The Figure 2 was revised by re-drawing the curves of fractional survival and adding SER(D0.5) data to support the radiosensitizer effects of IDO1 knockdown or IDO1 inhibitor. We also revised the Figure 4 simultaneously.
- In Fig. 3E, it is not surprising that inhibition of Notch1 activation leads to a reduction of NICD binding of any of its promoter binding sites. Perhaps the emphasis should be on the detection of NICD bound to the IDO1 promoter.
Responses:
We apologized about the unclear descriptions of Fig. 3E. The antibody used for ChIP was anti-Notch1 antibody and the detection region of DNA was the putative RBPJ/CSL binding site at +140 of IDO1 promoter. The results truly demonstrated the binding of NICD to IDO1 promoter. We revised the descriptions about Fig. 3E in the maintext as following: “With chromatin immunoprecipitation (ChIP) method by using an anti-Notch1 antibody to pull-down chromatins followed by detection of IDO1 promoter with qPCR analysis, we further found that the treatment of Ro-4929097 strongly suppressed the binding of NICD to the IDO1 promoter at the +140 site, one of the putative RBPJ/CSL binding sites, in both HeLa or SiHa tumorspheres (Fig. 3E).”
- Also in Fig. 3E and Fig. 5C, there is no rationale for why specific putative binding sites were selected. Data for other sites should be included.
Responses:
Although we have designed the primer sets for detection the other putative sites of RPBJ/CSL within IDO1 promoter or AhR/ARNT sites within Notch1 promoter, the Ct values of these primer sets were ranged from 30 to 32 which makes the difficulty for ChIP use. However, the Fig. 3E or Fig. 5C clearly demonstrated the binding of NICD to the +140 site of IDO1 promoter or ARNT to the -309 site of Notch1 promoter, respectively.
- The methods do not provide sufficient information to review the number of biological/technical replicates and statistical methods. Methods section should include statement of scientific rigor and statistical methods. Comparison of individual data points in colony formation assays by t test is insufficient. See Franken, N., Rodermond, H., Stap, J. et al. Clonogenic assay of cells in vitro. Nat Protoc 1, 2315–2319 (2006).
Responses:
We thank the suggestions from the reviewer to make us to revise the calculation of our data of clonogenic assay as the responses of point 3 and the sensitizer enhancement ratios were calculated according to the report from Naumann et al (PloS one 2017, 12, e0180940) to demonstrate the sensitization effects of IDO1 shRNAs or INCB-024360, the IDO1 inhibitor, in Figure 2 or the knockdown of Notch1 in Figure 4. The biological/technical replicates were added in the figure legends. We also add a section of Statistical analysis in the Materials and Methods section as 4.12 in this revised manuscript.

Reviewer 2 Report
The manuscript is clearly presented overall . The authors suggested that silencing of IDO1 suppresses Notch 1 activation, while inhibition of Notch 1 decreases the expression of IDO1. Downregulation of IDO1 increases radiosensitivity of cervical cancer. And kynurenune decreases the radiosensitivity on the other hand. The authors suggested that ISO1 inhibitor can be a radiosensitizer for cervical cancer patients.
- The panel annotation in Figure 1 does not match with the legends. Please revise. For the Western blot analysis, there is no N in the figure.
- Is the IDO1 inhibitor specific?
- There are typos on lines 159 and 172.
- In line 225, authors mentioned "HeLa or SiHa" but in the figure there is only SiHa data presented.
- Is BMI1 also a well-known factor for cancer stemness? It was not mentioned anywhere in the manuscript but only in Figure 7.
- How is the dosage of IDO1 inhibitor injection be determined?
Author Response
The manuscript is clearly presented overall . The authors suggested that silencing of IDO1 suppresses Notch 1 activation, while inhibition of Notch 1 decreases the expression of IDO1. Downregulation of IDO1 increases radiosensitivity of cervical cancer. And kynurenune decreases the radiosensitivity on the other hand. The authors suggested that ISO1 inhibitor can be a radiosensitizer for cervical cancer patients.
Responses:
We thank the positive feedbacks from the reviewer.
-The panel annotation in Figure 1 does not match with the legends. Please revise. For the Western blot analysis, there is no N in the figure.
Responses:
We apologized for the unmatched legends and the “N” was revised as “2D” in the revised manuscript.
-Is the IDO1 inhibitor specific?
Responses:
As mentioned by the article from George C Prendergast et al. (Cancer Res. 2017. 77(24):6795-6811.), the INCB-024360 is a highly selective inhibitor for the IDO1 enzyme without affecting the related (IDO2 or TDO2) or unrelated (Cyp P450 enzymes) enzymes. This reference has been added as ref 20 in this revised manuscript. We also add some descriptions about INCB-024360 as “…INCB-024360, a small molecule inhibitor of IDO1 with a highly specificity and a lead agent for clinical evaluations [20],…” (page 3, line 122-123).
- There are typos on lines 159 and 172.
Responses:
We thank the comments from the reviewer and the typos were collected.
- In line 225, authors mentioned "HeLa or SiHa" but in the figure there is only SiHa data presented.
Responses:
We thank the comments from the reviewer and the maintext has been revised as deleting HeLa from line 243 of revised manuscript.
- Is BMI1 also a well-known factor for cancer stemness? It was not mentioned anywhere in the manuscript but only in Figure 7.
Responses:
BMI1 is also considered as the factors for cancer stemness but we did not observe any change by kynurenine treatment. The descriptions have been revised as “….,, as well as the increased expression of Oct4 or Sox2 but not BMI1 (Fig. 7C), which all belong to the well-known cancer stemness genes [29].” (Page 9, line 246-247.).
- How is the dosage of IDO1 inhibitor injection be determined?
Responses:
The dosage of INCB-024360 for in vivo use was referenced from a report from Koblish et al (Mol Cancer Ther. 2010. 9(2):489-98.). This reference has been added to the 4.10 of Materials and Methods section as ref 39.

Round 2
Reviewer 1 Report
The authors have addressed all of my concerns. Only minor line/copy editing is necessary. Examples below.
1) Suggest better labels for Figure 1A.
2) Figure panel lettering system for Figure 1 are not aligned with figure legend.
3) Notch is misspelled on line 144, 160, 169, 184, 211, and 345.
Author Response
Responses to Reiverwer1
The authors have addressed all of my concerns. Only minor line/copy editing is necessary. Examples below.
Responses:
We thank the positive feedback from the reviewer.
1) Suggest better labels for Figure 1A.
Responses:
We apologized for the unclear labeling for Fig. 1A and it has been revised by adding the labels for indicating normal (green color) or tumor (red color) samples.
2) Figure panel lettering system for Figure 1 are not aligned with figure legend.
Responses:
We apologized for the mistakes of lettering in the figure legend of Figure 1 and it has been corrected in this revised manuscript.
3) Notch is misspelled on line 144, 160, 169, 184, 211, and 345.
Responses:
We apologized for the typos and they have been corrected in this revised manuscript.